# A diminutive new basilosaurid whale reveals the trajectory of the cetacean life histories during the Eocene

Mohammed S. Antar [1,2✉], Abdullah S. Gohar [1,3], Heba El-Desouky [1,4], Erik R. Seiffert [5], Sanaa El-Sayed[1,6], Alexander G. Claxton[7] & Hesham M. Sallam [1,3]

Soon after whales originated from small terrestrial artiodactyl ancestors, basal stem forms (archaeocetes) came to inhabit more specialized aquatic ecologies and underwent a tremendous adaptive radiation that culminated in the adoption of a fully aquatic lifestyle. This adaptive strategy is first documented by the geographically widespread extinct family Basilosauridae. Here we report a new basilosaurid genus and species, *Tutcetus rayanensis*, from the middle Eocene of Fayum, Egypt. This new whale is not only the smallest known basilosaurid, but it is also one of the oldest records of this family from Africa. *Tutcetus* allows us to further test hypotheses regarding basilosaurids' early success in the aquatic ecosystem, which lasted into the latest Eocene, and their ability to outcompete amphibious stem whales and opportunistically adapt to new niches after they completely severed their ties to the land. *Tutcetus* also significantly expands the size range of the basilosaurids and reveals new details about their life histories, phylogeny, and paleobiogeography.

[1] Mansoura University Vertebrate Paleontology Center (MUVP), Mansoura University, Mansoura 35516, Egypt. [2] Nature Conservation Sector, Egyptian Environmental Affairs Agency, Cairo 11728, Egypt. [3] Institute of Global Health and Human Ecology (I-GHHE), School of Sciences and Engineering, American University in Cairo, New Cairo 11835, Egypt. [4] Department of Geology, Faculty of Science, Mansoura University, Mansoura 35516, Egypt. [5] Department of Integrative Anatomical Sciences, Keck School of Medicine of USC, University of Southern California, Los Angeles, CA 90033, USA. [6] Department of Earth and Environmental Sciences, University of Michigan, Ann Arbor, MI 48109, USA. [7] Department of Anatomy and Cell Biology, Oklahoma State University Center for Health Sciences, Tulsa, OK 74107, USA. ✉email: m.s.antar@mans.edu.eg

Soon after whales originated in the late early Eocene from small quadrupedal terrestrial artiodactyl ancestors in south Asia[1–3], semi-aquatic archaeocete whales rapidly dispersed westward to North Africa[4,5], West Africa[6], North America[7,8], and South America[9] during the early middle Eocene. As the late Eocene (Priabonian) progressed, the fully aquatic basilosaurid archaeocetes had replaced amphibious archaeocetes[10] and were the most abundant whales and ultimately extended into the geographic ranges occupied by modern cetaceans[11].

Basilosauridae was the first family of archaeocete whales known to science[12]. Basilosaurids were cosmopolitan, anatomically derived, and fully aquatic archaeocete whales that are thought to be close to the ancestry of extant (or crown) cetaceans[10,13]. Basilosaurids are characterized by the loss of the maxillary third molar, modification of their forelimbs into flippers, a substantially reduced innominate which lacks any bony connection to the vertebral column, an increased number of posterior thoracic/lumbar vertebrae, and the development of paddle-like tails[14]. They range in size from around 4 m for *Saghacetus osiris* (Priabonian) to around 18 m for *Basilosaurus cetoides* (Bartonian to early Priabonian)[10,11,14].

Basilosaurids are the best known archaeocetes of the African Paleogene[15]. Not only were the vast majority of their fossil remains discovered in Egypt[14–16], but the most thoroughly documented basilosaurids, including the first fully aquatic cetaceans and early tail-powered swimming cetaceans, were discovered in Egypt's Fayum Depression, which is home to the Wadi El-Hitan World Heritage Site, one of the world's most productive fossil whale sites[17]. The German naturalist Georg Schweinfurth discovered the first fossil cetaceans from the eastern hemisphere (and indeed the entire African continent) in Egypt in 1879[18]. These fossils included a variety of isolated vertebrae of archaeocete whales from the Birket Qarun Formation on the Geziret El Qarn island of Qarun Lake in the Fayum Depression, Egypt[19]. Since then, the richness of cetacean fossils in the African (mainly Egypt) fossil record has substantially shaped our understanding of early whale evolution[14–17]. This has led to a greater understanding of the diversity, anatomy, behavior, and adaptations of archaeocete whales[4,5].

Here, we report on a new basilosaurid whale, *Tutcetus rayanensis*, gen. et sp. nov., from the middle Eocene (early Bartonian) Sath El-Hadid Formation of the Fayum Depression, Egypt. The holotype specimen, Mansoura University Vertebrate Paleontology Center (MUVP) 501, is an incomplete skull with mandibles, the hyoid apparatus, and the atlas vertebra of a small-sized subadult basilosaurid whale in an indurated limestone block. The new whale is the smallest basilosaurid known to date and is estimated to have been around 2.5 m in length and about 187 kg in body mass. It is not only the smallest basilosaurid whale yet discovered, but it is also one of the oldest records worldwide. This intriguing specimen markedly expands the size range of the basilosaurids and demonstrates that whales achieved considerable disparity during the middle Eocene.

## Results

### Systematic paleontology. Mammalia Linnaeus, 1758

Cetacea Brisson, 1762
Pelagiceti Uhen, 2008
Basilosauridae Cope, 1868
*Tutcetus*, new genus

*Etymology*. A combination of *Tut*, for the ancient Egyptian Pharoah Tutankhamun, commonly known as King Tut, who unexpectedly died in his 18th year, and *cetus*, Greek for a whale. Genus name is used in reference to the subadult status and the diminutive size of the type specimen.

*Type species*. *Tutcetus rayanensis*, new genus and species.

*Generic diagnosis*. The specimen was assigned to the family Basilosauridae based on the presence of multiple accessory cusps on the cheek teeth and well-developed pterygoid sinuses around the auditory region. *Tutcetus rayanensis* differs from other basilosaurids by its diminutive size (possibly the smallest known basilosaurid); it further differs in having a maxilla that abuts most of the lateral sides of the nasal, leaving only a small anterior portion of the nasal to articulate with the ascending process of the premaxilla; the number of mesial and distal accessory cusps on the upper and lower premolars (mainly two mesially and three distally). Furthermore, the premolars of *Tutcetus rayanensis* are more gracile than those of any other known basilosaurid and have extremely smooth enamel, and the fourth premolar ($P^4_4$) is the largest tooth in both the upper and lower jaws. *Tutcetus rayanensis* lacks replacement of the first premolar. The Supplementary Information provides a more detailed diagnosis (Supplementary Note 1).

*Tutcetus rayanensis*, new species

*Etymology*. A combination of "*Rayan*," in reference to the Wadi El-Rayan Area, the locality of the holotype, and "*ensis*" (Gr., N.L. masc. adj.).

*Species diagnosis*. As for genus.

*Holotype*. Mansoura University Vertebrate Paleontology Center (MUVP) 501 (see Supplementary Table 1 and Supplementary Figs. 1, 2), an associated partial skeleton consisting of a partial cranium, dentaries, hyoid apparatus, and the atlas vertebra of a subadult basilosaurid whale (Figs. 1, 2, Supplementary Figs. 3–12 and Supplementary Tables 2–7).

*Type locality and horizon*. Wadi El-Rayan valley (40 km northeast of Wadi El-Hitan World Heritage Site) of the Fayum Depression in the Western Desert of Egypt (Supplementary Figs. 1, 2). The Sath El-Hadid Formation, early Bartonian, late middle Eocene (ca. 41 Ma)[20]. The Supplementary Information provides a more detailed description of the locality and stratigraphy (Supplementary Note 2).

**Description**. The holotype cranium of *Tutcetus rayanensis* (MUVP 501) is preserved from the premaxilla to the occipitals, including nasal, frontal, presphenoid, sphenoid, and much of the braincase (Supplementary Fig. 3). However, the palatal surface is mostly destroyed (see Supplementary Results). A part of the premaxilla is detached from the rest of the cranium and retains the upper left second incisor ($I^2$) that is embedded in its body and projects anteriorly and buccally. The posterior part of the left maxilla is preserved and bears the alveoli for the upper posterior teeth. The maxilla bounds most of the lateral side of the nasal, leaving only a small anterior portion of the nasal free to articulate with the ascending process of the premaxilla. The nasal is exposed in ventral view, from its anterior edge to the straight fold of the dorsal nasal concha around the dorsal nasal meatus. The vomer is triangular in cross-section and separated dorsally into two crescent-shaped wings that form both the lateral walls and the posterior roof of the narial cavity. Posteriorly, at the level of the mid-frontal, the vomer contacts and slightly covers some portions of a robust presphenoid. Anteriorly, the basisphenoid has a pentagonal cross-section. The posterolateral edge of the basisphenoid articulates with the anterior lamina of the pterygoid. In MUVP 501, the frontal is complete, with a small preorbital process and a thicker postorbital process that limits the anterior

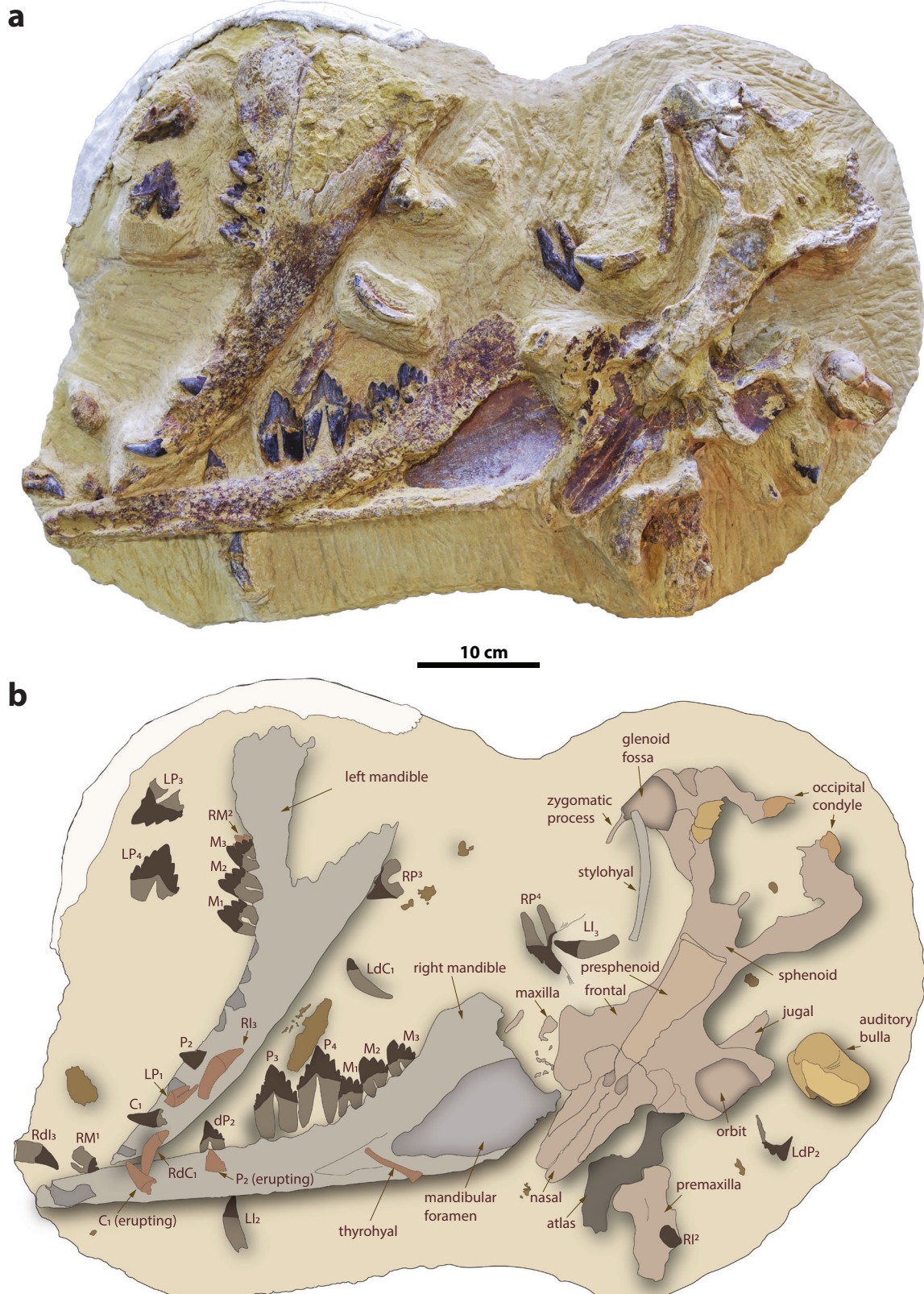

**Fig. 1 *Tutcetus rayanensis* (MUVP 501, holotype).** Photograph (**a**) and corresponding explanatory line drawing (**b**) of the block containing the holotype specimen of *T. rayanensis* (MUVP 501).

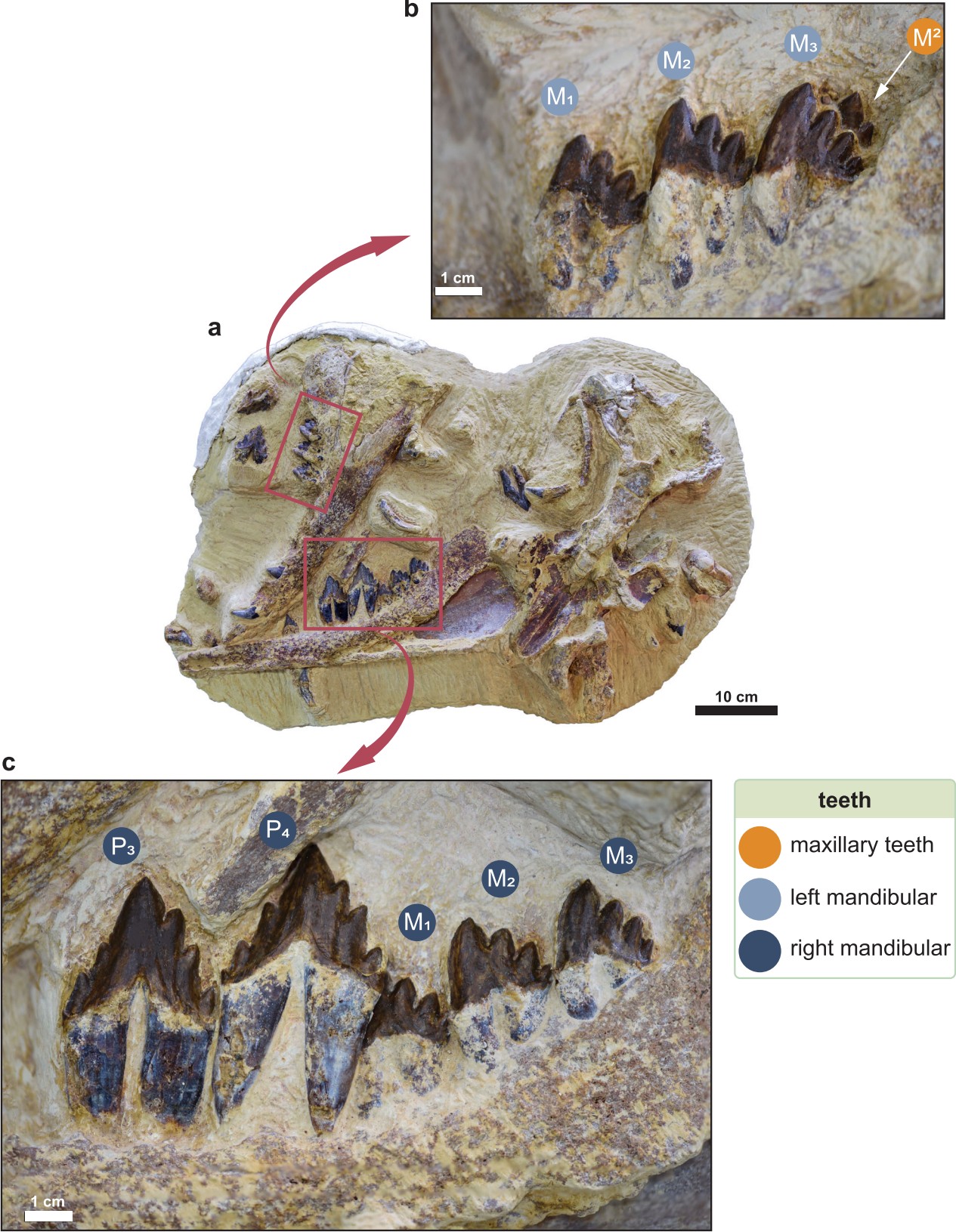

**Fig. 2 Tooth morphology of *Tutcetus rayanensis* (MUVP 501, holotype). a** The block containing the holotype specimen of *T. rayanensis* (MUVP 501).
**b** Close-up of the posterior lower teeth in the left (**b**) and right (**c**) dentaries of *Tutcetus rayanensis* (MUVP 501, holotype).

edge of a small temporal fossa. The lateral edge of the frontal is concave between these two processes on the ventral surface, resulting in a semicircular depression that serves as the orbit's roof. Medial to the orbit, a pair of sinuses are enclosed by the frontals. Anteriorly, the parietal sutures with the frontal just posterior to the level of the postorbital process of the frontal. The frontoparietal suture protrudes laterally and extends ventroposteriorly until it reaches the alisphenoid. Posteriorly, the parietal meets the supraoccipital and forms the nuchal crest. A parietal foramen lies anterior to the dorsalmost margin of the parietosquamosal suture. Dorsomedially, left and right parietals articulate along the midline to form a strong and prominent sagittal crest. The occipital condyle of MUVP 501 measures around 20 mm in width and is better preserved on the left side of the skull. The foramen magnum is oval in shape, measuring 33.5 mm in width. The nuchal surface of the supraoccipital rises 83.7 mm above the foramen magnum to the apex of the nuchal crest. The nuchal crest is rectangular when viewed posteriorly. On the left side of the skull, the squamosal projects laterally forming the zygomatic process. The left zygomatic process is thickened anterior to the glenoid fossa and then becomes laterally compressed more anteriorly. Just posterior to the postglenoid process is the external auditory meatus.

The tympanic bulla of *Tutcetus* (Supplementary Fig. 4) is similar to those of other basilosaurids[14,21] and it is ovoid to rectangular in shape (55 mm long and 43 mm wide), being expanded transversely. *Tutcetus* has a slightly convex medial edge on its tympanic bulla. While the ventral surface of the bulla is smooth and convex, it presents lateral and medial eminences separated by the interprominential notch near the posterior edge of the bulla. The tympanic cavity on the dorsal side separates the transversely thicker involucrum on the medal side from the very thin outer lip on the lateral side. The outer lip of the bulla preserves a well-developed subtubular sigmoid process, which is incomplete in MUVP 501. The sigmoid process originates ventrolaterally, projects dorsolaterally then is twisted posteromedially on the dorsal side of the bulla. The involucrum presents a low keel that runs anteroposteriorly along the medial side of the bulla, but it is more pronounced posteromedially.

Both dentaries of the holotype are preserved and almost complete (Fig. 1). Although the left and right dentaries are unfused, the mandibular symphysis is obscured. The lateral surface of the dentary is convex, with the convexity being most pronounced in the posterior third of its length. The medial surface of the dentary is gently concave. The coronoid process is broad and sweeps up in a gentle arc that peaks just slightly higher than the main cusp of $M_3$. The mandibular foramen is small, reaching a height of about the same as the height of the mandible at $M_3$. The shape of the medial border of the mandibular foramen is triangular, similar to that of *Dorudon atrox* (SMNS 11417a). The right condyle is elongated mediolaterally and placed on a shallowly excavated process on the medial side.

The right deciduous lower third incisor ($dI_3$) is single-rooted and has a conical crown that is laterally compressed and curves slightly lingually and distally (Supplementary Fig. 5). Near the base of the crown, the enamel is crenulated, with most of the crenulations visible on the lingual surface. The mesial and distal sides of the root are separated by a small groove on the lingual side of the root. There is a resorption cavity at the base of the root for the crown of the permanent $I_3$. Both the left and right deciduous lower canines ($dC_1$) are preserved in MUVP 501 (Fig. 1 and Supplementary Fig. 11). $dC_1$ is notably smaller than the lower incisors, with a rounded and less buccolingually compressed crown that is relatively shorter and smaller at the base than any other incisor (Supplementary Table 3). The distal portion of the root of $dC_1$ is hollowed out to accommodate the

crown of $C_1$. The left second lower deciduous premolar ($dP_2$) is isolated, while the right $dP_2$ is in its original position in the mandible above the right $P_2$ (Supplementary Figs. 5, 6, 10, 11). Both the left and right $dP_2$ are buccolingually compressed, with a prominent main cusp, no mesial accessory cusps, and two distal accessory cusps. No cingulum is visible on the labial or lingual surfaces of $dP_2$, and the enamel is smooth and wrinkle-free.

The premaxilla of MUVP 501 retains a root for the single-rooted upper left second incisor ($I^2$) in its alveolus (Supplementary Fig. 6). Only the right upper posterior premolars ($P^{3-4}$) are preserved (Supplementary Fig. 6). $P^3$ is subequal in size to $P^4$ and both are double-rooted, buccolingually compressed, have a strong primary cusp above the root division, and both have a distolingual expansion on the distal root and crown. The cingulum is well-developed lingually and forms small mesial and distal denticles. $P^3$ has no accessory cusps on the mesial edge and two accessory cusps on the distal edge. On $P^4$, the mesial edge presents three accessory cusps and the distal edge bears two tiny accessory cusps (the higher cusp being much larger than the lower one) and many tiny blunt tubercles along the distal edge (Supplementary Fig. 6), similar to that on $P^4$ of *Chrysocetus* (CCNHM 119) and the protocetid *Georgiacetus* (GSM 350). On $P^4$, the distal accessory cusps are substantially smaller than the mesial ones. The upper molars ($M^{1-2}$) are double-rooted, labiolingually compressed, and have a lingually enlarged distal root (Supplementary Fig. 9). *Tutcetus*, as revealed by the CT scan, lacks the upper third molar ($M^3$). $M^{1-2}$ are subequal in size, $M^1$ being slightly longer than $M^2$. Their triangle-shaped crown is mesiodistally longer than it is high and is noticeably lower in labial and lingual views than those of the premolars. In $M^1$, there are three mesial and two distal accessory denticles on the edges of the crown. $M^2$ has one mesial and two distal accessory denticles on the edges of the crown. Denticles become smaller from mesial to distal and from the apex to the base of the crown.

MUVP 501 retains the left lower third and second incisors ($I_{2-3}$), as well as the right $I_3$, and all are detached from their alveoli (Supplementary Figs. 5, 6). The left $I_{2-3}$ are visible, while the right $I_3$ is concealed beneath the left mandible, as determined by a computed tomography (CT) scan (Supplementary Figs. 9, 12). All incisors are single-rooted, subequal in size, and do not have a cingulum or accessory denticles. The crown of the incisors is conical, smooth, bent distally, and labiolingually gently compressed. The roots of the incisors are slightly curved with no vertical ridge. On both dentaries, the lower canine ($C_1$) is single-rooted and still erupting (erupted more on the left dentary; Supplementary Figs. 10, 11). It has a conical smooth crown that is slightly bent distally and labiolingually compressed with no accessory cusps or cingulum. Only one alveolus is visible at the level of the first lower premolar ($P_1$) (Supplementary Fig. 5), even though a CT investigation of the left mandible revealed that the lower $P_1$ is double-rooted and fully developed (Supplementary Figs. 10, 12). Although they are double-rooted, the roots are coalescent, subequal in size, and tightly compressed, yet they only occupy a single alveolus without an intervening bony septum. On the left dentary, more than half of the crown of $P_2$ erupted, and roughly one-third of the crown of the right $P_2$ erupted. $P_2$ is double-rooted, buccolingually compressed, and has only one little accessory cusp, which projects straight up, on the distal edge. The third ($P_3$) and fourth ($P_4$) lower premolars are preserved on both sides of the mandible. $P_4$ is slightly larger than $P_3$ (Supplementary Table 3) and both are double-rooted and buccolingually compressed (Fig. 2 and Supplementary Fig. 6). In lateral view, the crowns of $P_3$ and $P_4$ are straight and triangular, with a prominent, robust main cusp above the root division that lacks wear. The enamel is very smooth. There are two accessory cusps mesially and three distally on the edges of $P_{3-4}$. On both teeth, the

highest distal accessory cusp is closer to the main cusp than the highest mesial accessory cusp. Overall, the accessory cusps on the mesial side are smaller than those on the distal side, and all of them are substantially smaller than the main cusp. The cingulum of P$_{3-4}$ is well-developed lingually, forming a denticle on both the mesial and distal edges. In P$_3$, the distal root is longer than the mesial root, and they are roughly parallel with nearly equal spacing between them. In P$_4$, the size of the mesial and distal roots is roughly equal and divergent downwards. There is no diastema separating P$_3$ and P$_4$, as well as between the posterior premolars and the following molars. Both the left and right sets of lower molars (M$_{1-3}$) are still in situ in the holotype dentaries. All lower molars are double-rooted, small, triangular in labial view, and labiolingually compressed, with a primary cusp and three accessory denticles only on the distal edge of each molar. There is a gradual increase in the size of the primary cusp through M$_{1-3}$. The size of the distal accessory denticles decreases from subequal to the apical cusp to substantially smaller in the most distal accessory denticle. The crown in M$_{1-3}$ is an asymmetrical triangle, with no wear signs, and the apex is slightly shifted distally. The mesial edge is labiolingually wider than the distal edge. The enamel is smooth and lacks wrinkling. The mesial edge of the crown forms a sharp ridge mesiolingually, and a reentrant groove mesiolabially, which houses the most distal accessory denticle of the preceding tooth. The mesial root is transversely inflated with a labial expansion. Lingually, an incipiently developed cingulum is observed in the distal region, where it forms the base of the smallest accessory denticle on the distal edge of the crown. There is no diastema separating the lower molars.

The hyoid apparatus preserved in MUVP 501 includes stylohyal and thyrohyal elements (Supplementary Figs. 7, 10). The stylohyal has a smooth, long, cylindrical shaft with an oval to rounded cross-section, as opposed to a more rounded cross-section at the proximal and distal ends. The shaft is also slightly convex laterally with 10 degrees of divergence in both dorsal and ventral views. The distal end of the stylohyal has a rough surface, indicating that it most likely would have contacted cartilage. The thyrohyal element was identified under the right mandible via CT scanning of the holotype-bearing block. According to the 3D volume rendering of the resulting CT scanning, it is more robust than the stylohyal, as it is thicker and more rounded in cross-section with an almost straight thyrohyal body. This straight thyrohyal body differs from the strongly curved thyrohyal body of *Pontogeneus peruvianus* (MNHN.F.PRU10).

The atlas vertebra (C1) is made up of dorsoventrally thinner dorsal and much thicker ventral arches that enclose the neural canal (Supplementary Fig. 8). While the ventral arch presents two highly concave articular foveae cranially, it has a broad convex articular surface caudally. The transverse process of the atlas vertebra of *Tutcetus* lacks the distinctive narrow transverse process of *Saghacetus osiris* (UM 97550, UM 100140a). There is a 5 mm diameter vertebroarterial foramen perforating the base of the transverse process. The neural arch is dorsally convex, posteriorly oriented, and has a lateral vertebral foramen that opens into the neural canal. The neural canal is roughly oval to circular in shape, measuring 32 mm dorsoventrally and 33 mm transversely. The articular surface surrounds most of the neural canal on the medial side. C1 has a maximum length of 45 mm at the level of transverse processes, a minimum length of 23 mm separating the anterior and posterior articular facets, and a maximum width of roughly 129 mm.

**Phylogenetic relationships**. Our Bayesian tip-dating analysis (=BTD, Fig. 3, Supplementary Methods; Supplementary Results and Supplementary Figs. 13, 14) recovered a clade of georgiacetine protocetids (*Natchitochia, Georgiacetus, Babiacetus, Tupelocetus,* and *Aegicetus*) as the most crownward of the protocetid

clades and is the sister taxon of a moderately-supported (PP = 0.65) clade herein termed Pelagiceti, which contains all basilosaurids and Neoceti (Mysticeti and Odontoceti). Within the georgiacetine clade, *Aegicetus* and *Tupelocetus* were identified as the latest surviving protocetids. Within the Pelagiceti clade, the BTD analysis recovered Basilosauridae as paraphyletic, with *Eocetus* as the sister taxon of all other sampled members of Pelagiceti. The lineage that gave rise to *Eocetus* is estimated to have split off from all other members of Pelagiceti around 45 Ma, or around the start of the middle Eocene (i.e., during the middle Lutetian substage). All other basilosaurids (aside from *Eocetus*) and neocetes are included in a moderately-supported (PP = 0.67) clade that arises from the next-most crownward divergence from the cetacean stem lineage. This clade is estimated to have arisen ~44.4 Ma.

Among core basilosaurids, a weakly-supported (PP = 0.30) early-diverging basal clade of middle Eocene (Bartonian) basilosaurids from Africa (i.e., *Tutcetus, Chrysocetus*), North America (*Chrysocetus*), and South America (*Ocucajea*) was recovered. Within this clade, herein termed *Tutcetus*-clade, a moderately-supported (PP = 0.70) clade contains *Tutcetus* as the sister taxon of the South American *Ocucajea*, with the North American and African *Chrysocetus* being more stemward and the sister taxon of *Ocucajea* + *Tutcetus*. *Tutcetus*-clade was recovered as the sister taxon of a moderately-supported (PP = 0.62) clade that is comprised of all the late Eocene basilosaurids investigated. Within this clade, *Zygorhiza* is the sister taxon of all other late Eocene basilosaurids, which includes an *Ancalecetus-Saghacetus* clade (PP = 0.68) and a (*Basilosaurus* (*Dorudon, Pontogeneus*)) clade (PP = 0.65). The (*Dorudon, Pontogeneus*) clade is weakly-supported (PP = 0.47).

A more crownward clade of basilosaurids weakly (PP = 0.25) includes the Bartonian South American *Supayacetus* alongside other Bartonian basilosaurids from Europe (*Pachycetus paulsonii*), North America (*Pachycetus wardii*) and Africa (*Antaecetus aithai*). Within this clade, herein termed Pachycetinae, the *Antaecetus-Pachycetus* clade is well-supported (PP = 0.84) with *Pachycetus paulsonii* being more basal and a sister taxon of the moderately-supported (PP = 0.64) clade that includes *Pachycetus wardii* + *Antaecetus aithai*. The Pachycetinae clade has estimated divergence dates ranging from 43 to 42 Ma and it is the sister taxon to later, more derived neocetes (Fig. 3).

**Ancestral State Reconstructions (ASRs)**. Bayesian ancestral state reconstructions (ASRs; Supplementary Results, Supplementary Figs. 15–20 and Supplementary Tables 8–16) were obtained for all characters. ASRs for skull length (Character 3; Supplementary Fig. 15 and Supplementary Table 8) reveal that a long skull (skull length >800% of condylar breadth) is the likely ancestral state of archaeocetes and all later cetaceans. The moderate skull length (skull length 700–800% of condylar breadth) in protocetids such as *Artiocetus* and *Georgiacetus*, as well as the basilosaurids *Tutcetus, Zygorhiza* and *Dorudon* is a derived trait. A short skull (skull length <700% of condylar breadth) arose twice, first in the basal middle Eocene protocetid *Rhodocetus* and later in the late Oligocene mysticete *Mammalodon*. ASRs further suggest that the common ancestors of *Tutcetus*-clade (Node #86) and Neoceti (Node #78) probably had moderate skull length. A large supraorbital process (Character 19; Supplementary Fig. 17 and Supplementary Table 11) is a unifying feature of the clade containing the *Tutcetus* + *Ocucajea* (Node #87). The absence of a supraorbital process is the probable ancestral state of the earliest stem whale families (Pakicetidae, Ambulocetidae, and Remingtonocetidae), while the supraorbital process seen in protocetids, basilosaurids,

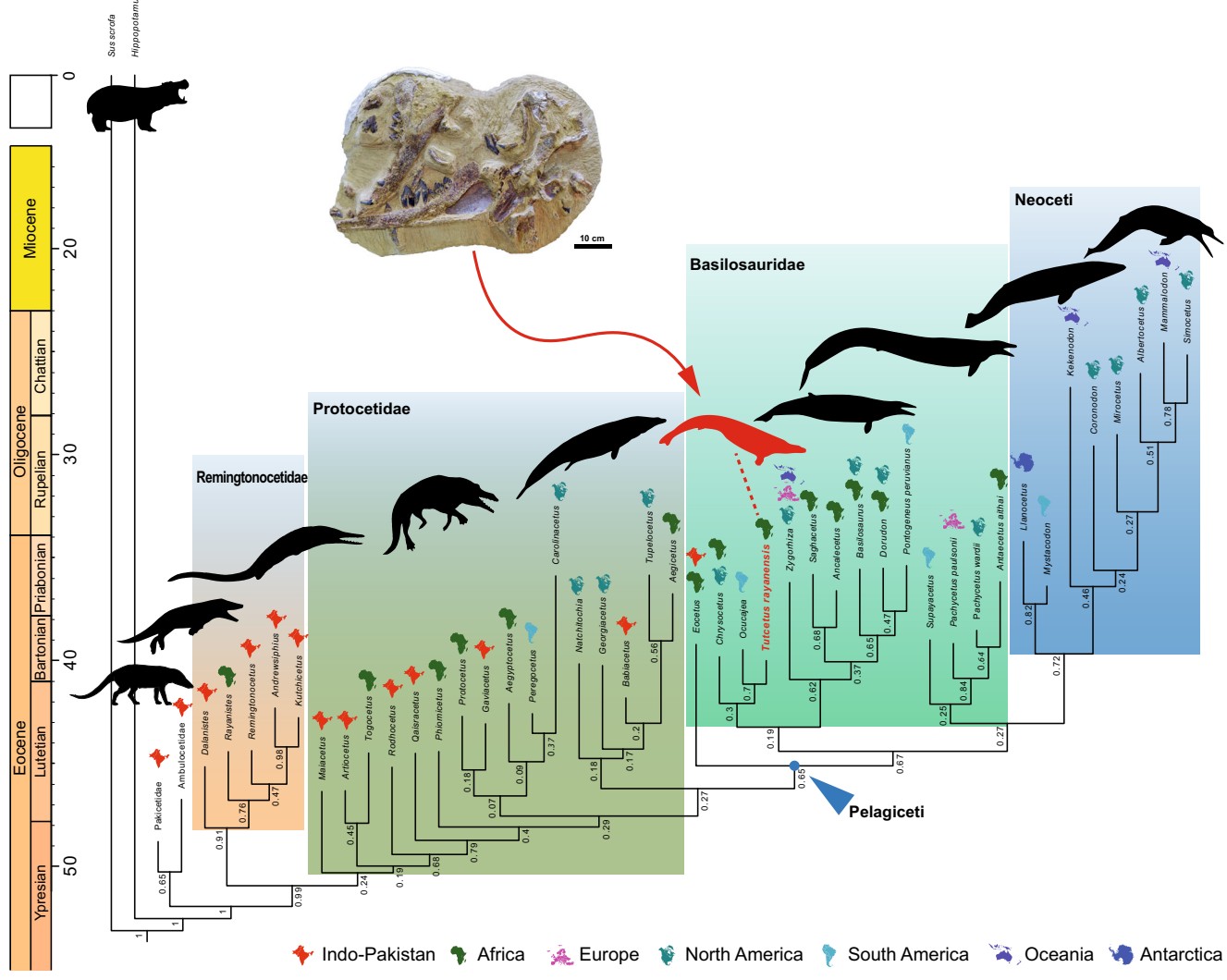

**Fig. 3 Phylogenetic position of *Tutcetus rayanensis* (MUVP 501, holotype).** Allcompat consensus tree from the Bayesian tip-dating analysis of the 195-character matrix in MrBayes 3.2.5 with the implementation of the FBD prior. Numerical values to the left of nodes represent posterior probabilities (PPs). Icons above taxon names reflect the geographic location of each fossil. A photograph of the block containing the holotype specimen of *Tutcetus rayanensis* (MUVP 501) is shown above its corresponding position (indicated red arrow).

and Neoceti is likely a derived trait. Furthermore, ASRs show that the large supraorbital process seen in the clade containing *Tutcetus* + *Ocucajea* (Node #87) is the most probable ancestral state along a basal part of the cetacean stem lineage, and was retained by *Rodhocetus*, *Qaisracetus*, and *Georgiacetus*. Analysis of another trait, the angle between the posterior edge of the postorbital process and the sagittal crest (Character 23; Supplementary Fig. 17 and Supplementary Table 12), revealed that the right angle between the posterior edge of the postorbital process and the sagittal crest, seen in the clade containing *Tutcetus* + *Ocucajea* (Node #87) first evolved in archaeocete along the stem lineage that gave rise to all protocetids (Node #57). ASRs further show that the acute angle between the posterior edge of the postorbital process and the sagittal crest obtained by some basilosaurids and some Neoceti first appeared in the common ancestor of all basilosaurids and Neoceti (Node #72). ASRs for the accessory denticles on P$^3$ (Character 101; Supplementary Fig. 19 and Supplementary Table 13) reveal that the equal number of denticles on both edges of the tooth is a derived trait that first appeared along the branch leading to *Eocetus*, basilosaurids and Neoceti (Node #72). Therefore, the addition of more denticles on the distal edge of the tooth, as

seen in *Tutcetus* + *Ocucajea* (Node #87), and in *Zygorhiza*, is a derived trait. This derived trait arose again in the late Oligocene in *Kekenodon*. The development of a P$^3$ subequal in length to P$^4$ (Character 113; Supplementary Fig. 15 and Supplementary Table 14) is reconstructed as having evolved convergently, first in the common ancestor of the remingtonocetids (Node #53) and second in the common ancestor of the *Tutcetus*-clade (Node #86). This trait also arose later in the common ancestor of the earliest (late Eocene) toothed baleen whales, *Mystacodon* and *Llanocetus* (Node #84). Smooth premolar enamel (Character 118; Supplementary Fig. 20 and Supplementary Table 15) also evolved convergently in the *Tutcetus*-clade (Node #86), the Bartonian pachycetine *Antaecetus aithai* (Node #77), and in the Oligocene odontocete *Simocetus rayi*.

## Discussion

Although sexual maturity cannot be directly determined from fossils, the age at which the permanent dentition has completely erupted is closely linked to the age of sexual maturity in most mammals[14]. The ontogenetic age structure of *Tutcetus rayanensis* was estimated by comparing its dental eruption sequence

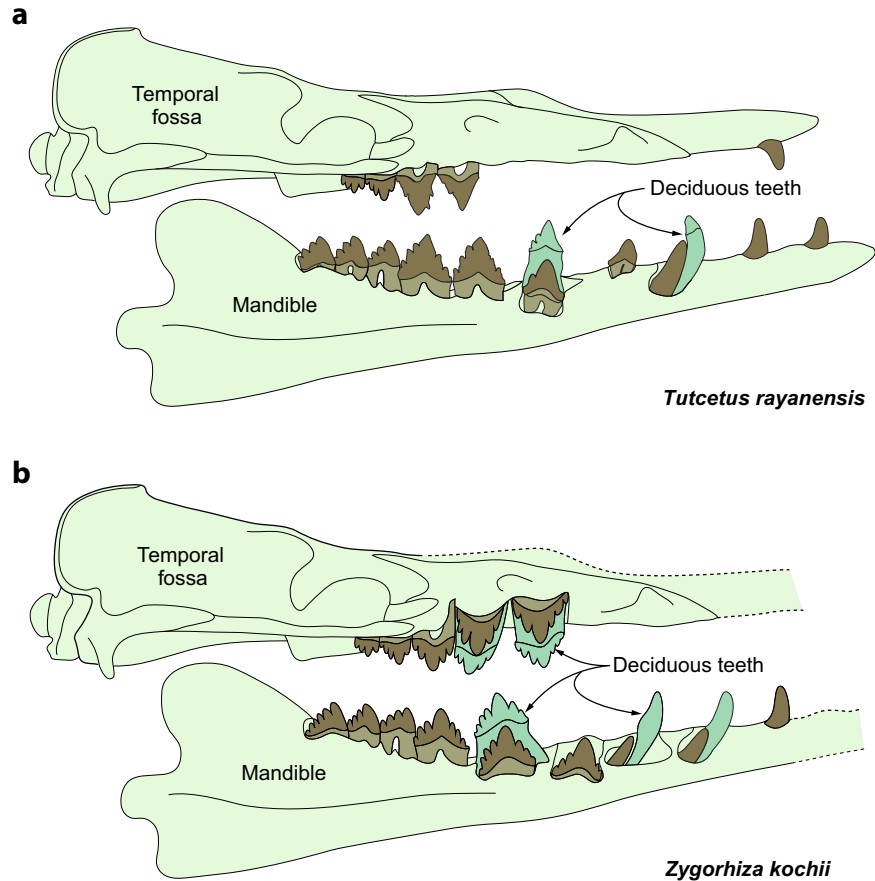

**Fig. 4 Comparison of tooth replacement in the basilosaurid whales.** *Tutcetus rayanensis* (**a**) and *Zygorhiza kochii* (**b**).

to those of several *Dorudon atrox* and *Zygorhiza kochii* individuals in different ontogenetic stages[14] within a series of thirteen age classes (Supplementary Fig. 21 and Supplementary Table 4).

All observable cranial sutures of the holotype specimen of *Tutcetus rayanensis* are notably fused (Supplementary Tables 5, 7). The only postcranial vertebral element in association with the holotype of *Tutcetus rayanensis* is an atlas (C1) vertebra, in which the left and right halves of the neural arch are fused together. Based on the stage of the dental eruption, closure of the cranial sutures, and the epiphyseal fusion stages (Supplementary Table 5), *Tutcetus rayanensis* appears to have been at a more advanced stage in its life history than was the oldest known juvenile specimen of *Dorudon atrox* (UM 94795, in which $M_{1-3}$ are in place, $P_4$ is partially erupted, and $dP_3$ is in place). The *Tutcetus* holotype is older than the holotype specimen of *Chrysocetus healyorum* (SCSM 87.195), and the specimen of *Zygorhiza kochii* that had almost completed its dental eruption (USNM 16639). The eruption sequence presented by *T. rayanensis* suggests that its pattern of dental development differs from the general trend of the dental eruption schedule in dorudontine basilosaurids (Fig. 4 and Supplementary Table 6). The eruption sequence in *T. rayanensis* suggests that, in contrast to *Dorudon* and *Zygorhiza*[22], there is no $P_1$ replacement, which is a usual condition for mammals, and suggests that *Tutcetus* may either retain a functional $dP_1$ into adulthood that is never replaced, or erupt a successional $P_1$ without a deciduous precursor. It further suggests that the second premolar ($P^2_2$) is the last tooth to erupt (Supplementary Table 6). There is a noticeable contralateral (left versus right) differential in the eruption order of the permanent teeth in *Tutcetus*, indicating that the left side of the mandible had an earlier tooth eruption than the right side of the mandible.

Furthermore, regular diphyodonty in *Tutcetus* follows the same trajectory as other archaeocetes, which differs from suggested monophyodonty or very early tooth replacement in *Chrysocetus healyorum* (SCSM 87.195) and also differs from monophyodonty in modern cetaceans. Based on the fusion of cranial sutures, advanced stage of teeth eruption, and epiphyseal fusion in the dorsal and ventral arches of the atlas, and by comparison of those with the ontogenetic stage of skeletal fusion and eruption sequences in dorudontine basilosaurids (Supplementary Tables 5–7), *T. rayanensis* is clearly an advanced subadult, i.e., near adulthood, individual.

The dental replacement pattern is a fundamental key for inferring the pace of life histories over a broad range of fossil taxa[23]. Replacement teeth usually erupt quite early and concurrently with the molars in mammals that have slower life histories, longer lifespans, and slow maturation[24,25]. In their dental eruption sequence, mammals with early-erupting molars (i.e., *Tutcetus*) have relatively fast life histories, shorter lifespans, and rapid onset of sexual maturity. As a result, its early molar eruption and lack of $P_1$ replacement suggest rapid dental development in *Tutcetus*. This rapid dental development coupled with its exceptionally small size (188 kg), indicate that *Tutcetus* likely had a precocial lifestyle and a fast pace of life history; that is, it lived its entire life rapidly and died at a younger age than did larger and later occurring basilosaurids and neocetes.

In order to mate and give birth to their young, adult whales typically migrate to warm, shallow, tropical waters (breeding grounds)[26]. The depositional environment of the strata where fossils of *Tutcetus* have been unearthed likely offered ideal calving grounds for cetaceans. There is a high (~63%) mortality rate in young juveniles of *Dorudon atrox*[14] and no specimens

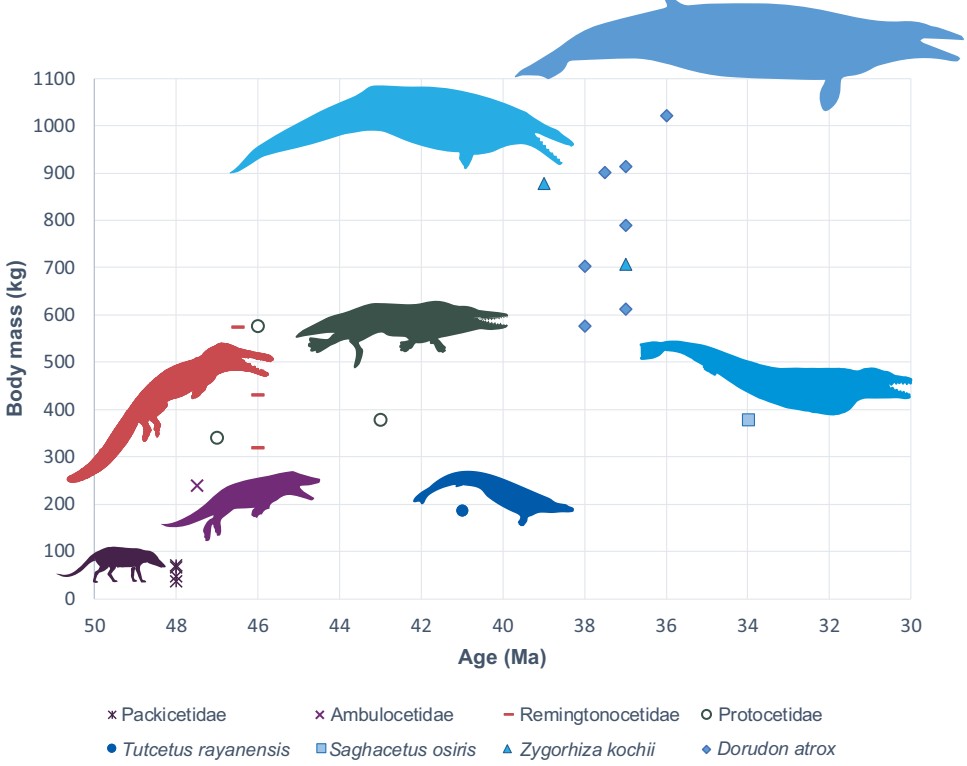

**Fig. 5 Evolution of the size of the archaeocete whales over the Eocene.** Scatter plot illustrating the relationship between body mass (kg) and age (Ma) for several archaeocete whales estimated using the breadth across the occipital condyles. Note that archaeocete whales exhibit a general trend of increasing body mass with the geologic age, while *Tutcetus rayanensis*, in contrast, has a very small body mass (187 kg).

have been found whose anterior premolars, incisors, or canines are still erupting. In contrast, the only known individual of *Tutcetus* died while in the process of erupting its anterior premolars, incisors, and canines. This may indicate that young juveniles of the original *Tutcetus* population have a relatively reduced mortality rate, but this hypothesis can only be tested through the recovery of more specimens. Since similar patterns of mortality are anticipated in populations of mammals in which females give birth once a year[27,28], the rate of mortality in young juveniles of the individuals of *Tutcetus* and *D. atrox* suggests that these species were likely to have given birth to singleton offspring each year.

Prior to the appearance of the basilosaurids, early cetaceans were relatively small[1–3] still tethered to the land, and dependent on hindfoot propulsion[9] as observed in phocid seals[29]. By the middle Eocene, basilosaurids had abandoned their ties to the land for a purely marine existence[17]. The relationship between body size and environmental temperature varies inversely over a wide variety of taxa[30–32]; when contrasting closely related species and distinct populations within a species, animals tend to evolve large body masses in colder climates. At times ranging from seasonal cycles to paleoclimate transitions, body size also declines when temperatures rise[32]. Aquatic taxa experience body-size reductions with warming more severely than terrestrial taxa[32].

*Tutcetus* has been unearthed from a stratum that is Bartonian in age (late middle Eocene). The relatively small size of *Tutcetus* (188 kg) (Fig. 5 and Supplementary Table 17) is either a primitive retention, or alternatively might be connected in some way to global warming during the late Lutetian thermal maximum (LLTM)[33]. The middle Eocene climatic optimum (MECO)[34], or the brief cooler period between LLTM and MECO events, is likely associated in some manner with the later and fast size increase of basilosaurids of the middle Bartonian-Priabonian (Supplementary Fig. 22).

Senescence and mortality are more prevalent in warmer climates[25], encouraging early maturation at a smaller size (i.e., *Tutcetus*) and a second generation can be produced with greater speed. It is hypothesized that resource rivalry had a considerable impact on body shape (e.g., elongation in Basilosaurinae), whereas changes in body size (reduced body size in *Tutcetus*) were mostly impacted by environmental temperature. Therefore, in an environment of limited resources, individuals who were proportionally more elongated and possibly faster swimmers had an advantage. In order to better understand competitive hierarchies and species interactions, body shape responses may therefore be a sign of changes in resource availability. Therefore, we urge that body shape and size be considered in tandem when examining the ecological effects of global warming on cetaceans.

The discovery of *Tutcetus* (Supplementary Figs. 23, 24) and the Eocene fossil record of archaeocete whales from Egypt[4,5] and Africa[6,15,35] are both generally notable and pertinent because they contribute to the emerging scenario of early cetacean evolution and shed light on the group's paleobiogeography. *Tutcetus*, coupled with other archaeocete records from Africa[4–6,35] South America[9,36], and North America[37], show that the transition of cetaceans from semiaquatic to fully aquatic probably happened in the (sub)tropics. In our BTD analyses, *Tutcetus* was consistently recovered as a close relative of other Bartonian dorudontines. This, in turn, suggests that *Tutcetus* holds considerable palaeobiogeographic implications. In the first tip-dating analysis, the *Tutcetus*-clade is estimated to have diverged from the late Eocene basilosaurids clade at or immediately before the estimated date of the earliest trans-Atlantic dispersal of cetaceans roughly ~45.7 and ~43.3 million years ago[5]. Recovering the Bartonian *Tutcetus*-clade raises the possibility that the South American (Peru) *Ocucajea picklingi*[36] and the North American (USA) *Chrysocetus healyorum*[38] may be the result of a single dispersal event from Africa to South and North America. Although

poorly constrained, this temporal window overlaps with other estimates for multiple dispersals across the Tethys from Asia into Africa (anthropoids, hystricognaths rodents)[39], and indeed when Sirenia first appears in the Western Hemisphere[40]. According to the fossil record from Antarctica[41] and Peru[36], basilosaurids most likely achieved a rapid spread over the Southern Hemisphere, reaching high latitudes by the middle Eocene. It is possible that Eocene basilosaurids from Antarctica[41], which resembled those from New Zealand[42] more than those from Peru[36], followed a dispersal route through the marine environments of the Australasia region (Supplementary Fig. 22).

## Methods

**Fieldwork and computed tomography**. The study was conducted in line with the Egyptian Environmental Affairs Agency's (EEAA) mission and remit. The EEAA, responsible for monitoring the Wadi El Rayan Protected Area, condoned this study. The holotype-bearing block was entirely subjected to CT imaging to supplement observations of the morphology on the external surface. The block was scanned helically using a Philips Incisive CT Scanner at the Urology and Nephrology Center at Mansoura University, Egypt. The overview scans were made with a field of view of 500 mm, a tube voltage of 140 kV, a tube current of 272 mA, and a slice thickness of 0.8 mm. The scanning resulted in 1949 slices with a pixel matrix of 512 × 512, a field of view of 500 mm, and a 0.2 mm interval between slices. Observations on image slices and three-dimensional (3D) visualization were done using the software package Aviso 4.1.2 (Visage Imaging Inc., Chelmsford, MA) in Oklahoma State University Center for Health Sciences (Oklahoma, USA).

**3D laser scanning**. The 3D laser scanning of fossils was performed with the hand-held Artec Leo 3D scanner with a 3D point accuracy of up to 0.1 mm and a 3D resolution of up to 0.5 mm, while its 3D accuracy over 1 m is as small as 0.03%. Its furthest range of linear field of view is 838 mm × 488 mm, and its largest angular field of view is 38.5° × 23°. The scanned data from the Artec Leo was transformed into the post-processing software Artec Studio 17.

**Phylogenetic analyses**. To determine the relationships of *Tutcetus rayanensis* within Basilosauridae and Archaeoceti, a Bayesian tip-dating (BTD) analysis was utilized using a revised archaeocete-dominated matrix of ref. [5] (Supplementary Data 1) and the pelagicete-dominated matrix of ref. [21] (Supplementary Data 2). The Supplementary Information describes the changes that were made to the datasets used for the phylogenetic analysis (Supplementary Methods). The BTD analysis employed the fossilized birth-death (FBD) process for the archaeocete-dominated matrix, which contained 47 taxa and 195 characters, and several characters were treated as ordered and constrained extant *Sus* and *Hippopotamus* as consecutive outgroups using hard constraints. Age priors for most taxa were based on Paleogene stage boundaries provided in the GTS 2020[43], except in cases where ages could be further constrained by other geochronological or biostratigraphic evidence. Settings for other priors were as follows: nodeagepr=calibrated; clockvarpr=igr; igrvarpr=exp(10); samplestrat=fossiltip; extinctionpr=beta(1,1); fossilizationpr=beta(1,1); speciationpr=exp(10); sampleprob=0.006 (based on a count of 345 non-camelid artiodactyls, two of which were sampled); and clockratepr=normal (0.01,100). The treeagepr prior was set as truncatednormal(52.04,52.05,1.0), based on the maximum possible age of Pakicetidae. The Markov Chain Monte Carlo (MCMC) analysis was run for 50 million generations in the MPI version of MrBayes 3.2.7[44], with sampling every 1000 generations, nchains=4, swapfreq=1, and temp=0.10. The first 25% of the samples were discarded as burnin. The average standard deviation of split frequencies (ASDSF) was 0.005058 in the final generation, and the minimum effective sample size (min-ESS) for all parameters was 1192.64, providing strong evidence for convergence. The BTD analysis of the pelagicete matrix, which contained 31 taxa and 101 characters, employed the same approach as that which was used for the analysis of the archaeocete matrix, with the exception that 1) a null (i.e., all "?") outgroup with a fixed age prior of 0 was used alongside Pakicetidae in order to facilitate calculations of divergence dates relative to the present, and 2) sampleprob was set to 0.00001. For that analysis, the ASDSF in the final generation was 0.002549, and the min-ESS for all parameters was 2689.65, clearly indicating convergence. Trees were summarized using the allcompat (majority-rule plus compatible groups) consensus.

For ASRs on the allcompat trees derived from BTD analyses, we used MBASR v. 2022.11.06[45] with n.samples set to 5000. Synapomorphies for the clades of interest reported in the main text (transitions between fixed and/or polymorphic states) were identified using the cutoffs provided in the output of the plots by MBASR (Supplementary Data 3). ASRs for all characters on the archaeocete and pelagicete trees are provided as supporting information.

**Body length and mass estimates**. The body mass of *Tutcetus*, as inferred by width across the occipital condyles[46], is estimated to be 187.1 kg (Supplementary

Table 17), which results in an estimated body length of 2.5 m. Body mass was calculated using the following equation from Waugh and Thewissen[46], which relates the mass (in kilograms) to the width across the occipital condyles (OCW in mm):

$$Log_{10}(BM) = 3.135 \times Log_{10}(OCW) - 3.575$$
$$Log_{10}(BM) = 2.2721009604999 \tag{1}$$
$$\text{Body Mass(Kg)} = \mathbf{187.1}$$

Where body mass is in kilograms, and occipital condyle width (OCW) in millimeters; the equation produces an adjusted $R^2 = 0.87$, a standard error of ±0.232 for the slope, and ±0.52 for the intercept, with a lambda of 0.942. This result was then entered into a separate equation that relates body mass to body length $(L)$[47]:

$$\log_{10} BM = 2.799 \cdot \log_{10} L - 4.464 \tag{2}$$

$$Log_{10}(L) = \frac{Log_{10}(BM) + 4.464}{2.799}$$
$$Log_{10}(L) = 2.40660984655 \tag{3}$$
$$Length(cm) = \mathbf{255.04}$$

In order to evaluate the accuracy of our estimations for body length and mass, we utilized an additional equation for stem Odontoceti to calculate the body length of *Tutcetus*. This equation was derived from a linear regression analysis of measurements taken from living cetaceans[48,49]. This equation estimates the total body length (cm) from the bizygomatic width of the skull in cm:

$$\text{Log (total body length)} = 0.92 \cdot (\text{Log (BZW)-1.72}) + 2.68$$
$$= 0.92 \times (\text{Log}(26.1) - 1.72) + 2.68$$
$$= 2.4009092668 \tag{4}$$
$$\mathbf{Body\ length} = 251.7 \mathbf{cm}$$

This result was again entered into Eq. (2) which relates body mass to body length:

$$\text{Log BM} = 2.799 \cdot \text{Log } SL - 4.464$$
$$\text{Log BM} = 2.799 \cdot \text{Log}(251.71509873) - 4.464 \tag{5}$$
$$\mathbf{BM} = \mathbf{180.4 kg}$$

Therefore, based on the cranial measurements, *Tutcetus*' body mass is estimated to be between 180.4 and 187.1 kg, and its body length is between 251.7 and 255.04 cm.

**Nomenclatural acts**. This published work and the nomenclatural acts it contains have been registered in ZooBank, the proposed online registration system for the International Code of Zoological Nomenclature. The ZooBank LSIDs (Life Science Identifiers) can be resolved and the associated information viewed through any standard web browser by appending the LSID to the prefix "http://zoobank.org/". The LSIDs for this publication are: FF3C4CB9-A4B2-4522-A15E-B732FC1B507B for the genus; 3D0004C9-0CCD-490D-B63F-9A6E6E40AE38 for the species.

**Reporting summary**. Further information on research design is available in the Nature Portfolio Reporting Summary linked to this article.

## Data availability

All data supporting the findings of this study are available within the paper and its Supplementary Information files. Additionally, the data that support the findings of this study have been deposited in figshare and are available in three supplementary datasets: Supplementary Data 1 (Archaeocete-dominated matrix of *Tutcetus* in Nexus format), Supplementary Data 2 (Pelagicete-dominated matrix of *Tutcetus* in Nexus format), and Supplementary Data 3 (results from the ASRs on the allcompat trees derived from Bayesian tip-dating analyses in a Zip file). The figshare repository can be accessed at https://doi.org/10.6084/m9.figshare.22811183. The holotype specimen of *Tutcetus* (MUVP 501) is housed in the Mansoura University Vertebrate Paleontology Center (MUVP), Mansoura University, Egypt.

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

## Acknowledgements

We are grateful to the Urology and Nephrology Center at Mansoura University, Egypt, for performing the computed tomographic scanning. We thank S. Heritage for his assistance with the ASR analysis. We acknowledge M. Talaat, the Egyptian Environmental Affairs Agency, for accessing the Wadi El-Hitan and Wadi El Rayan Protected Areas. We would like to thank the park rangers and the Wadi El-Hitan and Wadi El Rayan Protected Areas staff for their diligent help. We are grateful to members of the Sallam Lab for their enthusiastic support and help with preparing and photographing the specimen.

## Author contributions

M.S.A. excavated the specimens. M.S.A. and A.S.G. performed the anatomical description, carried out comparative work, and drafted the manuscript. Additionally, M.S.A. and A.S.G. modified the character matrix, scored phylogenetic matrices, generated the figures, and wrote the supplementary material. H.E. and S.E. engaged in discussions about the results and contributed to certain aspects of the Supplementary Information. E.R.S. conducted phylogenetic analyses, while M.S.A., A.S.G., and E.R.S. collaborated in describing and interpreting these analyses. A.G.C. managed the processing of the CT data. H.M.S. designed and supervised the project. All authors contributed to the interpretation of the results and discussions and contributed to the editing of the manuscript.

## Competing interests

The authors declare no competing interests.
