## [Peer Review File · Communications Biology]

Reviewers' comments:

Reviewer #1 (Remarks to the Author):

This is a well written and informative manuscript. The new specimen clearly represents a new species and is especially interesting because of its early age and discovery in a formation that has not previously been known to include archaeocete whale fossils. I have a few minor comments that are indicated in the manuscript.

The only ones I will note here are this. I suggest the authors seriously consider not naming a new subfamily. I think is wise to be cautious regarding the naming of new higher taxa, and this is the first time this clade has been identified. In addition, the reported phylogeny includes an unusual feature showing pachycetines most closely related to Neoceti. There are features of this group that suggest to me that they are basal among basilosaurids, so it immediately jumped out to me as unusual. If the authors retain this phylogeny, I think it would be worth explaining this grouping more, given how unusual it is.

Reviewer #2 (Remarks to the Author):

Review of

A diminutive new basilosaurid whale reveals the trajectory of the cetacean life histories during the Eocene

This manuscript represents an exceptionally well-conceived and well-written piece of science dealing with the description of a new genus of an archaeocete whale and its implications in archaeocete phylogeny, life strategies, and paleobiogeography. Illustrations are very well-made; methods are clearly described; the diagnoses are clear and detailed; the descriptions are good; the phylogenetic analysis is well-conceived and realized. Results are interesting to a wide audience of evolutionary biologists and experts in cetacean paleontology and evolution. I think that this manuscript is worth publication after minor revisions.

My principal concern is the lack of a Conclusions section in which the main results should be summarized. I encourage the authors to add a Conclusions section in order to help the reader to understand their results in brief time. Abstract and Conclusions are very important because not all the readers are able to follow sophisticated anatomical descriptions and complex methodologies but they may be interested in understanding the impact of the specimen in our general knowledge of whale evolution.

Secondarily, I encourage the authors to add more comparisons with other archaeocete whales. Comparisons are rather scarce in the present manuscript but they are necessary to better understand the place of the specimen in archaeocete diversity and evolution. I understand that the authors added comparisons in the reconstructions of ancestral states but these may be uneasy to follow. I would like to see more comparisons about the atlas and the tympanic bulla, for example. These would be very useful for other students. Therefore, I ask the authors to add more comparisons in traditional way.

Additional minor points are summarized below.

Supplementary Figure S4. The authors write that they are illustrating the auditory region of the skull but, in reality, they are illustrating only the auditory bulla so I recommend them to replace region with bulla in the caption.

Supplementary Figure S8. A hypophysis is indicated in the atlas but the hypophysis is a neuroendocrine structure of the brain and not an osteological formation in the atlas. Probably, the authors are referring to the ventral tubercle which is present in the atlas of many mammals.

Response to Reviewer #1

	Referee comment	Reply
General Comment	This is a well written and informative manuscript. The new specimen clearly represents a new species and is especially interesting because of its early age and discovery in a formation that has not previously been known to include archaeocete whale fossils. I have a few minor comments that are indicated in the manuscript.	We would like to thank Reviewer #1 for his/her valuable comments on our manuscript. We have carefully reviewed and addressed each of these comments in order to further enhance the quality and clarity of our work.
Comment 1	The only ones I will note here are this. I suggest the authors seriously consider not naming a new subfamily. I think is wise to be cautious regarding the naming of new higher taxa, and this is the first time this clade has been identified.	We would like to clarify that our intention was not to propose or designate a new subfamily in this study. Instead, our aim was to conveniently refer to the clade containing Tutcetetus in the phylogenetic analysis throughout the manuscript. We initially used the term "Chrysocetinae clade" for this purpose. However, we understand that the terminology closely resembled the naming style of a subfamily, which could have led to confusion and been potentially misleading. To avoid misunderstandings, we changed the clade's name to " Tutcetetus -clade". This modification has been made consistently throughout both the manuscript [lines: 314, 317, 344, 372, 375, 485, 488] and the supplementary information.
Comment 2	In addition, the reported phylogeny includes an unusual feature showing pachycetines most closely related to Neoceti. There are features of this group that suggest to me that they are basal among basilosaurids, so it immediately jumped out to me as unusual. If the authors retain this phylogeny, I think it would be worth explaining this grouping more, given how unusual it is.	We consider your suggestion and have added a detailed explanation of this grouping in the supplementary information (Part 5). The explanation is as follows: “Our phylogenetic analysis of Dataset 1 demonstrates a notable affinity between pachycetines and Neoceti (Figure 3). However, the phylogenetic position of pachycetines remains somewhat uncertain due to the limited availability of well-preserved specimens. Pachycetines exhibit some basal traits among basilosaurids, such as relatively thicker posterior mandible walls and primitive auditory region anatomy. Nevertheless, their affinity with Neoceti may arise from

		specialized adaptations for utilizing low-frequency sounds, similar to those found in Miocene and extant baleen whales⁶². Our results highlight the shared anatomical features between pachycetines and both early and modern Neoceti⁶³. For instance, some pachycetines (e.g., GMTSNUK 2638) display an anteriorly inclined supraoccipital shield, an autapomorphy of Neoceti. Furthermore, the maxillae diverge from the typical basilosaurid morphology¹⁴, bearing resemblance to the earliest mysticete, Mystacodon selenensis⁶⁴⁻⁶⁵. The tooth row also terminates more anteriorly than in basilosaurids such as Pontogeneus peruvianus²¹ and Dorudon atrox¹⁴, with the alveolus for the rearmost tooth situated directly ventral to the lacrimal canal—similar to Mystacodon selenensis⁶⁴. These shared features offer persuasive evidence for an evolutionary relationship between pachycetines and Neoceti. However, additional, more comprehensive specimens are required to ascertain the relationship between these groups”.
--	--	--

Comments of Reviewer #1 reported in the annotated pdf

	Referee comment	Reply
Comment 3 Lines [83-87]	You neglected to include your new subfamily Chrysocetinae in here. It also needs a diagnosis and list of included taxa. Or, you could just drop it from the text and the Supplementary Information.	We would like to reiterate that our intention was not to name a new subfamily, so there is no need for a diagnosis or a list of included taxa. We kindly ask you to refer to our response to comment 1, where we have addressed this issue in more detail.
Comment 4 Line [124]	The supplementary information does not give much more detail on the locality. It would be nice to have some general coordinates, at least down to minutes or a single digit decimal degree.	We have added the coordinates of the holotype's location in the supplementary information (Part 1). The updated locality information is as follows: “The holotype of Tutcetis rayanensis (gen. et sp. nov.) was found in an indurated limestone block near the Upper Lake in the Wadi El-Rayan area (29°15' N, 30°25' E), a locality 40 km NE of the Wadi El-Hitan World Heritage Site in Fayum, Egypt”.

Comment 5 Lines [214-221]	Please clarify in here that this taxon lacks M3/. I figure it did, but a clear statement is needed.	Thank you for bringing this to our attention. We have updated the manuscript to include a clear statement that Tutcetetus lacks M³. Specifically, we have added the following sentence: "Tutcetetus, as revealed by the CT scan, lacks the upper third molar (M³)."
Comment 6 Lines [322-330]	This result that pachycetines are sister taxon to Neoceti is very unusual. There are several anatomical features of this group that suggests a more basal position among the basilosaurids.	We kindly request that you reference our response to comment 2, where we have addressed this issue in further detail. We recognize that the scarcity of well-preserved pachycetine specimens and the complexity of their morphological traits make it challenging to draw definitive conclusions about their phylogenetic position.

Response to Reviewer #2

Comment No.	Referee comment	Reply
General Comment	This manuscript represents and exceptionally well-conceived and well-written piece of science dealing with the description of a new genus of an archaeocete whale and its implications in archaeocete phylogeny, life strategies, and paleobiogeography. Illustrations are very well-made; methods are clearly described; the diagnoses are clear and detailed; the descriptions are good; the phylogenetic analysis is well-conceived and realized. Results are interesting to a wide audience of evolutionary biologists and experts in cetacean paleontology	We appreciate the careful review and comments from Reviewer #2. We have carefully reviewed and addressed each of these comments in order to further enhance the quality and clarity of our work.

	and evolution. I think that this manuscript is worth publication after minor revisions.	
Comment 1	My principal concern is the lack of a Conclusions section in which the main results should be summarized. I encourage the authors to add a Conclusions section in order to help the reader to understand their results in brief time. Abstract and Conclusions are very important because not all the readers are able to follow sophisticated anatomical descriptions and complex methodologies but they may be interested in understanding the impact of the specimen in our general knowledge of whale evolution.	We have prepared a conclusion section as per your recommendation. However, we would like to note that, based on our understanding, the journal's style guidelines do not specifically require a conclusion section, and it is ultimately up to the editor to decide whether to include it in the final manuscript. Nevertheless, we have written the conclusions section and have provided it below: “Conclusions We report a new basilosaurid whale, Tutcetus rayanensis (Supplementary Figs 24–25), found in the middle Eocene rocks (41 Ma) in the Western Desert of Egypt. Tutcetus is not only one of the oldest records of this family globally, but also the smallest known basilosaurid. This finding illuminates early whale evolution in Africa during the middle Eocene and provides crucial information on the breeding and calving grounds of early whales. Our detailed analyses of Tutcetus' teeth and bone elements allowed us to reconstruct its growth and development, offering clues into the life history of early whales. Tutcetus contributes to our understanding of the basilosaurids' early success in aquatic environments and their ability to outcompete amphibious stem whales and adapt opportunistically to new niches after completely severing ties to the land. The rapid dental development and small size (187 kg) of Tutcetus suggest a precocial lifestyle and a fast life history pace for early whales. This discovery plays a pivotal role in shaping our understanding of early cetacean evolution and the group's paleobiogeography, indicating that the transition from semiaquatic to fully aquatic cetaceans likely occurred in the (sub)tropics. Moreover, our Bayesian tip-dating analyses highlight the phylogenetic and palaeobiogeographic significance of Tutcetus, potentially pointing to a single dispersal event from Africa to South and North America. In conclusion, the discovery of Tutcetus offers valuable insights into early cetacean evolution, life history, and paleobiogeography, furthering our knowledge of these remarkable marine mammals”.

Comment 2	Secondarily, I encourage the authors to add more comparisons with other archaeocete whales. Comparisons are rather scarce in the present manuscript but they are necessary to better understand the place of the specimen in archaeocete diversity and evolution. I understand that the authors added comparisons in the reconstructions of ancestral states but these may be uneasy to follow. I would like to see more comparisons about the atlas and the tympanic bulla, for example. These would be very useful for other students. Therefore, I ask the authors to add more comparisons in traditional way.	We have considered your recommendation and have included a comprehensive Part 3 in the Supplementary Information that provides a detailed comparison with other archaeocete whales. Although we initially considered adding the comparison to the main text, we felt that due to the recommended word limit of the journal, it was best to include it in the Supplementary Information. The detailed comparison is as follows: “Tutcetetus differs from Pontogeneus peruvianus (MNHN.F.PRU10) and Supayacetus muizoni (MUSM 1465) in having equal sizes of the medial and lateral posterior prominences in the auditory bulla. Tutcetetus further differs from Dorudon atrox¹⁴, Pontogeneus peruvianus (MNHN.F.PRU10), Zygorhiza, Chrysocetus, and Saghacetus [similar to Ocucajea picklingi (MUSM 1442)] in having the posterior edge of the postorbital process oriented at approximately 90° to the sagittal crest and in having an anterodorsal orientation of the supraoccipital shield. Ocucajea picklingi (MUSM 1442) has a relatively wider supraorbital region than does Tutcetetus and further differs from Tutcetetus in having diastemata between its lower molars. Tutcetetus differs from Ancalocetus and Dorudon atrox (but not Zygorhiza kochii⁵⁵) in having convex articular facets on the atlas for the axis. Tutcetetus lacks the pinching of the nuchal crest just dorsal to the occipital crest that is seen in Saghacetus osiris⁵³. In contrast to Chrysocetus (SCSM 87.195) and Saghacetus osiris⁵³, which have a single-rooted P₁, the left dentary of MUV 501 preserves a double-rooted P₁, similar to Dorudon atrox¹⁴ and Zygorhiza kochii⁵⁵. Tutcetetus differs from Antaocetus³⁵ in being smaller, having more gracile teeth, and having a different configuration of the accessory cusps on premolars. Furthermore, the upper molars of Tutcetetus differ from those of Antaocetus³⁵ in lacking distolingual expansion. The skull of Tutcetetus further differs from Antaocetus³⁵ in having a relatively narrow constriction of the rostrum. In contrast to Saghacetus osiris⁵⁰, which has narrow transverse processes on C1, the atlas vertebra of MUV 501 shows wide transverse processes, similar to Dorudon atrox¹⁴ and Zygorhiza kochii⁵⁵.”
--	--

Comment 3	Supplementary Figure S4. The authors write that they are illustrating the auditory region of the skull but, in reality, they are illustrating only the auditory bulla so I recommend them to replace region with bulla in the caption.	We have made the necessary changes to the caption of Figure S4, substituting the term "auditory region" with "auditory bulla". The updated caption reads as follows: “ a, Block containing the holotype specimen of T. rayanensis (MUV 501) with the location of its right auditory bulla indicated by a red rectangle; b, Ventral view of a typical basilosaurid skull shows the location of the auditory bulla; digital model of the auditory bulla based on CT scans (c-e) and line drawing (f-h) of the right auditory bulla of T. rayanensis in dorsal (c, f), ventral (d, g), and posteromedial (e, h) views”.
Comment 4	Supplementary Figure S8. A hypophysis is indicated in the atlas but the hypophysis is a neuroendocrine structure of the brain and not an osteological formation in the atlas. Probably, the authors are referring to the ventral tubercle which is present in the atlas of many mammals.	We concur with your suggestion, and as a result, we have substituted the term "hypophysis" with "ventral tubercle," which is a more fitting osteological formation for the atlas. Figure S8: